# Emergent Challenges for Science sUAS Data Management: Fairness through Community Engagement and Best Practices Development

Jane Wyngaard [1,*,†] , Lindsay Barbieri [2,†] , Andrea Thomer [3] , Josip Adams [4] , Don Sullivan [5], Christopher Crosby [6] , Cynthia Parr [7] , Jens Klump [8] , Sudhir Raj Shrestha [9] and Tom Bell [10]

1   Center for Research Computing, University of Notre Dame, Notre Dame, IN 46556, USA
2   Rubenstein School of Environment and Natural Resources, Gund Institute for Environment, University of Vermont, Burlington, VT 05401, USA
3   School of Information, University of Michigan, Ann Arbor, MI 48109, USA
4   UAS Project Office, U.S. Geological Survey, Denver, CO 80225, USA
5   NASA Ames Research Center, Moffett Field, CA 94035-0001, USA
6   UNAVCO, Boulder, CO 80301, USA
7   Agricultural Research Service National Agricultural, Library United States Department of Agriculture, Beltsville, MD 20705, USA
8   Mineral Resources, Commonwealth Scientific and Industrial Research Organisation, Kensington, WA 6151, Australia
9   Esri Washington DC Regional Office, Vienna, VA 22182, USA
10   Earth Research Institute, University of California, Santa Barbara, CA 93106, USA
*   Correspondence: jwyngaar@nd.edu; Tel.: +1-574-631-5163
†   These authors contributed equally to this work.

**Abstract:** The use of small Unmanned Aircraft Systems (sUAS) as platforms for data capture has rapidly increased in recent years. However, while there has been significant investment in improving the aircraft, sensors, operations, and legislation infrastructure for such, little attention has been paid to supporting the management of the complex data capture pipeline sUAS involve. This paper reports on a four-year, community-based investigation into the tools, data practices, and challenges that currently exist for particularly researchers using sUAS as data capture platforms. The key results of this effort are: (1) sUAS captured data—as a set that is rapidly growing to include data in a wide range of Physical and Environmental Sciences, Engineering Disciplines, and many civil and commercial use cases—is characterized as both sharing many traits with traditional remote sensing data and also as exhibiting—as common across the spectrum of disciplines and use cases—novel characteristics that require novel data support infrastructure; and (2), given this characterization of sUAS data and its potential value in the identified wide variety of use case, we outline eight challenges that need to be addressed in order for the full value of sUAS captured data to be realized. We conclude that there would be significant value gained and costs saved across both commercial and academic sectors if the global sUAS user and data management communities were to address these challenges in the immediate to near future, so as to extract the maximal value of sUAS captured data for the lowest long-term effort and monetary cost.

**Keywords:** sUAS; drone; RPAS; UAV; data; management; FAIR; community; standards; practices

## 1. Introduction

Small Unmanned Aircraft Systems (sUAS)—also known as Remotely Piloted Aircraft Systems (RPAS), Unmanned Aerial Vehicles (UAV), or often colloquially as 'drones'—are rapidly becoming a ubiquitous tool for data collection across a wide range of private and public applications worldwide. This application space includes multiple academic fields (electrical, chemical, and civil engineering; environmental sciences; and others) for which sUAS are changing how and which data are captured. While this new platform shares much in common with traditional remote sensing techniques, the particular combination of varied spatiotemporal resolutions, operational practices, and wide spectrum of heterogeneous data being collected with sUAS has led to a unique set of data management challenges. Additionally, various global efforts and technological advances in the sphere of data management are opening unique opportunities for enhancing the potential of sUAS as an environmental sensing technology.

This paper compiles four years of extensive community engagement around the complexities, nuances, and importance of sUAS data management, and seeks to lays the motivations and foundations for future global sUAS user community engagements. We do so by: (1) outlining the potential value gains of normalising good data management practices for sUAS collected data, (2) detailing the unique complexities of sUAS data while pointing to analogous sectors and existing resources that might be leveraged, and (3) identifying key challenges and needs from the community in order to expand the value potential for sUAS data. Henceforth we will use "sUAS data" to refer to the primary research data captured on-board sUAS, rather than data relating solely to the sUAS platform itself. In many cases the former requires and therefore includes the latter.

To provide context for later sections, the remainder of this section outlines the current state of sUAS use in academia and the corresponding state of sUAS data management. Following which, Section 2 details the authors' engagement with the global community on this topic by: summarising what methods of community engagement were undertaken, including detailing which geographical regions and domains of expertise were included; and highlighting others working in this space and the resources that are currently available. Drawing on the outcomes of this engagement, Section 3 presents the core characteristics of sUAS captured data which inform the need for sUAS specific data management practices and infrastructure. Finally, Section 4 discusses the community distilled key challenges arising from Sections 2 and 3, before Section 5 concludes.

### 1.1. Current Use of sUAS in Research

The rapid adoption of sUAS for scientific data collection has been driven largely by the flexible functionality now possible due to key technological advances: lowered hardware costs, increased battery energy density, widespread sensor miniaturization, and the availability of sophisticated autopilot hardware and software. Lagging but globally following these technological advances, have been new aviation regulations in multiple countries [1] that facilitate the legal operation of sUAS. Thus, it is now possible and highly attractive for even small and modestly funded research teams to incorporate sUAS data into their investigations.

As platforms for scientific data collection, sUAS offer several functional advantages when compared with many traditional methods: (i) the ability to collect higher spatial and or temporal resolution data; (ii) a reduced impact on sensitive environments being monitored; (iii) lowered risks to workers and equipment involved in data collection in dangerous environments; (iv) a highly flexible platform from which a wide range of parameters might be monitored simultaneously; and (v) access to many data that what would otherwise be practically inaccessible, all (vi) often at a significantly lower cost than traditional methods might incur [2–4]. sUAS datasets are therefore generally parameter-rich and uniquely high-resolution datasets that consequently potentially offer unique and novel reuse value across multiple academic, commercial, governmental, and non profit use cases.

The value of these advantages to primary data users is evident in the number and domain variety of recently published peer-reviewed articles that include various terms for sUAS (see Figure 1).

This growth is paralleled by significant growth of the commercial sUAS sector (e.g., agriculture, mining, civil engineering, disaster response, cargo delivery, entertainment), with some forecasts estimating a market value of USD 100 billion in the next five to ten years [5–8]. The commercial market is further driving the rapid advancement of sUAS (flight platforms, sensor miniaturization, wireless telemetry, sophisticated autonomous navigation, operations, and legislation) to meet the needs of commercial sUAS use in: Agriculture, Mining, Civil Engineering and Infrastructure, Search and Rescue and Disaster Responses, Cargo and Data Delivery, Conservation, Entertainment, and many more use cases.

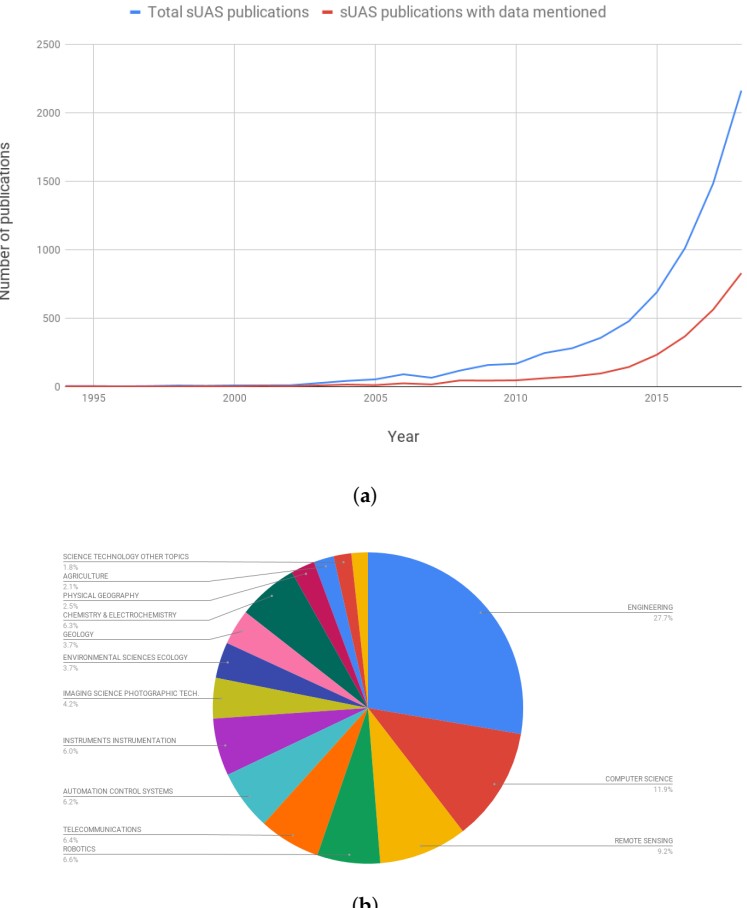

**Figure 1.** Graphs depicting growth and distribution of publications. In (**a**), as small Unmanned Aircraft Systems (sUAS) have become more prevalent as platforms for scientific data collection, there has been a corresponding increase in their prominence within the academic peer-reviewed literature. This graph shows this growth in blue, with the number of publications found in a Web of Science literature search on the topic of sUAS. By comparison, the number of sUAS publications that also referenced the management of data, is shown in red at a much lower rate. Search terms used in the search for papers referencing sUAS included: "unmanned aerial systems", "unmanned aerial system", "unmanned aircraft systems", "unmanned aircraft system", "unmanned aerial vehicle", "unmanned aircraft vehicle", "unmanned aerial vehicles", "unmanned aircraft vehicles", "remotely piloted aircraft system", "remotely piloted aircraft systems", "remotely piloted aerial systems", and "remotely piloted aerial system". For searches seeking both the terms used for sUAS and references to data management the following terms were included in relation to data: "data", "metadata", "data management", "data integration", "integrate data", "data fusion", "data standards", "data interoperability", "informatics", "data synthesis", and "data curation". Based on the same search terms, (**b**) categorizes the total number of publications returned, into the academic field under which they were published through 2018. (**a**) References to sUAS in Web of Science through 2018. (**b**) Percent publications in the 15 most prominent Web of Science research areas through 2018.

These advances are being made across and through novel and developing commercialization models as the industry evolves. As a result business models include the sale of both products and services, and both proprietary and open source sUAS solutions. For researchers, each of these models offers a variety of value trade-offs: from fully customizable and purpose-built solutions giving full access to all metadata, at the cost of development effort and time; through to less configurable but ready to fly platforms, or full data capture flight and post-processing analytics services. The latter generally also involves higher monetary costs, and provides less contextual information with their data but are easily and rapidly deployed.

### 1.2. Current sUAS Data Management

Research data management infrastructure and procedures have always been important but have become more complex and costly as the quantity of available data has significantly increased [9,10]. Why and how sUAS data management is critical to realising its full value is an outcome of this community-engaged work, discussed in more detail in Section 3. However, sUAS users who have attempted to publish their data are familiar with why it is also particularly challenging. A typical sUAS based project, for instance, will involve multiple stakeholders (e.g., scientists, engineer, pilot), technologies (e.g., sUAS, controllers, computers, software systems, sensors, paper notebooks), parameters (e.g., flight platform attitude, scientific sensor calibration date and processes, scientific parameters, comensual environmental conditions), and complex processes (e.g., data triage, data compression, data pre- and post-processing), many of which can impact the interpretation of the data. Capturing information about each of these disparate components is commonly necessary for initial data product generation and interpretation, and many are required for data publication and future reuse. Unfortunately, because scholarly and scientific sUAS users represent a relatively small user market with niche needs, the challenges of sUAS data management have not yet been widely addressed either by industry stakeholders.

As a result, individual researchers are developing their own ad hoc data management strategies. This is problematic in the long term for multiple reasons. First, this substantially adds to the learning curve of sUAS technologies: new-to-sUAS researchers must already navigate complex legal, technical, and institutional spaces, and developing a data management strategy from scratch further increases the required overhead. For researchers specifically seeking to take advantage of sUAS as a new and otherwise more affordable means of data capture, the economic and time costs of developing robust data management workflows can be prohibitive.

Second, the repeated reinvention of ad hoc data management workflows represents a significant amount of effort. Not only is this an inefficient use of finite research resources, but these idiosyncratic workflows pose a roadblock to the development of common tools and workflows. Without a collective articulation of sUAS data management best practices, there is no alternative even for those motivated to collaborate on the development of common better commercial and open source software and tools, exacerbating the issue as more ad hoc workflows are developed and used.

Third and finally, the lack of common data practices risks diminishing the trustworthiness and reproducibility of research based on sUAS data. Without shared data practices and methods of documenting workflows, sUAS data-based research is often plagued by poor or heterogeneous documentation, unknown or non-standard quality control methods, and methodological uncertainty. The current opacity of sUAS data workflows makes thorough scientific assessment and peer review difficult.

### 1.3. Opportunities for sUAS Data Management

The described landscape presents a problematic picture, yet the rapid growth of sUAS as a revolutionary sensor platform across multiple sectors has arrived at a highly opportune moment. Key developments and shifts in social, political, and particularly academic attitudes worldwide present a unique opportunity to the sUAS user community. Specifically, the coincidence of the following

present an opportunity: (i) the push for open science and FAIR (Findable Accessible Interoperable Reusable) [9] data, (ii) the corresponding maturing of data technologies, and (iii) the lack of momentum behind any substandard normalized practices and the minimal amount of legacy sUAS data currently available that would otherwise require significant effort to migrate or reprocess. The following elaborates on each of these opportunities.

### 1.3.1. The Push for Open Science and FAIRness

At the same time as sUAS are emerging as a standard tool for researchers, the broader research community is building momentum in actively moving towards normalising open science and FAIR data practices. This is evidenced by the wealth of work calling for better research practices [11–14]; the numerous calls for improving reproducibility and cross-disciplinary data use through better practices [15–18]; and the many non-academic calls for data sharing from a range of government bodies [19–21]. The significant traction that the FAIR nomenclature has gained—as a succinct framing of core good data management practices—further demonstrates this momentum [22–24].

### 1.3.2. The Corresponding Maturation of Data Technologies

As industry has moved to extract economic advantages from Big Data, the technologies required to manage, manipulate, and mine value in large and heterogeneous, datasets of mixed quality have significantly matured [25–27]. The breadth of associated tools available is extremely wide but a few high-visibility and relevant examples include; the growth in capabilities and use of cloud resources [28–31], Google's beta Dataset search engine [32] with the required enabling dataset schema, the international Earth and space science community's effort to develop standards that connect researchers, publishers and data repositories [33,34], and the increase in efforts to utilize Machine Learning tools on classical Big Data for a multitude of applications including the geosciences [35,36].

### 1.3.3. The Lack of Norms or Legacy sUAS Data

The lack of community accepted best practices for sUAS data management is both a challenge and an opportunity. As a new technology, researchers are still grappling with how best to use sUAS. This provides a window of opportunity within which: the "cost" of adopting new practices is minimal, and the net quantity of sUAS captured scientific data still relatively small, the cost of adopting new formats, metadata standards, calibration methods—and all of the other crucial components of data archival—will not be significantly added to by the need for backwards compatibility or mass re-ingestion and processing of previously captured data. This window, however, is closing rapidly, as researchers globally are creating all of these components for themselves in ad hoc and isolated manners, and rapidly accumulating data.

## 2. Materials and Methods

In light of the above, over the past four years (2014–2019) the authors have pursued a wide-ranging, largely volunteer-based, effort to engage with the growing community of researchers using sUAS for data collection on challenges of data management. Initially this involved looking to both the emerging sUAS science community and to the many mature analogous domains for applicable best practices, and included considering standards and conventions used by large scale government and research institutions using both sUAS and more traditional remote sensing technologies. This was followed by running multiple workshops and conference sessions with the aim of identifying key needs and available resources for sUAS data management. The progression of core engagement meetings involved are shown in Figure 2, at each of which we sought input from both academic and commercial sUAS users, suppliers, and developers and data management professionals. This process was largely driven by academic researcher needs, the perceived value opportunity, and community request and interest. As a result the focus has been on academic data more than commercial, however, both were consulted and across both the spectrum of fields of expertise engaged spans Engineering disciplines,

Earth and Environmental Sciences, Agricultural Sciences, library and information science, and the Humanities. In the following sections the key threads of this engagement are summarized and groups are highlighted for the purposes of directing interested parties to possible resources or potential starting points for future efforts.

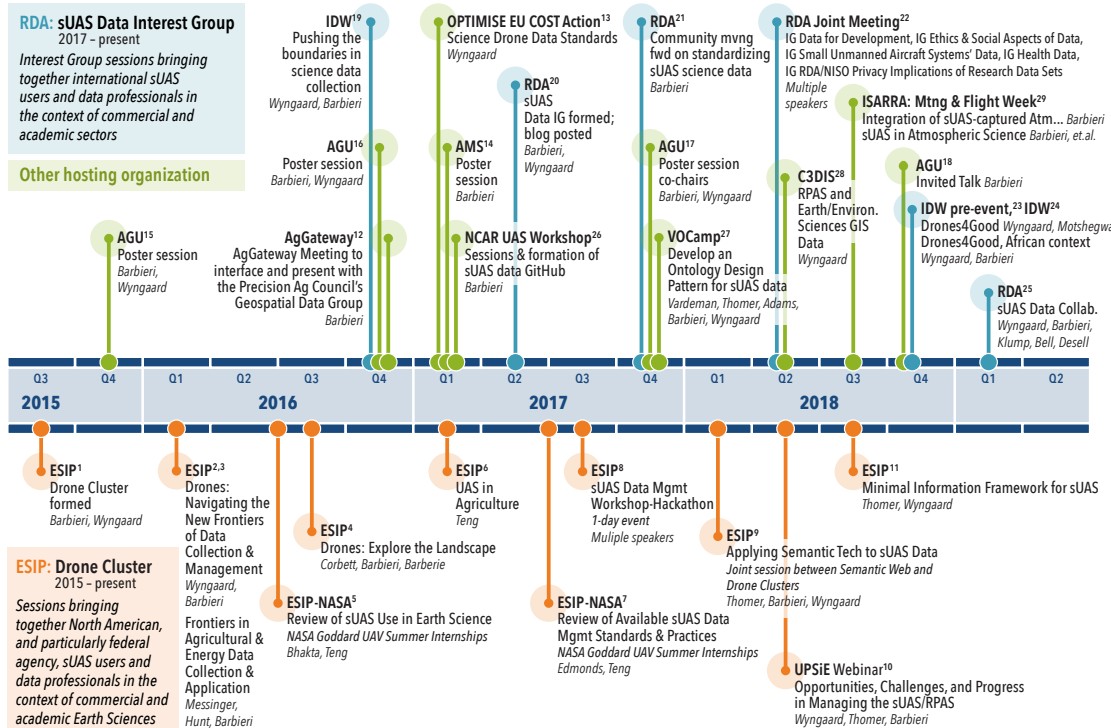

**Figure 2.** This timeline summarize the events the authors have used to engage with governmental organizations, commercial sUAS platform and tool providers, academic scientists, and both commercial and academic data professionals. Event references are available in Appendix A.

## 2.1. Community Engagement

### 2.1.1. Earth Science Information Partners Federation

Born out of a perceived need within the Earth Science Information Partners (ESIP) Federation a Drone Cluster [37] was initiated by authors Barbieri and Wyngaard in 2015. ESIP is *"an open, networked community that brings together science, data and information technology practitioners. ESIP is supported by NASA, NOAA, USGS, OGC, and 110+ member organizations"* [38]. Since then this cluster has run multiple sessions at ESIP meetings, hosted interns, and produced prototyping projects [39]. At the 2017 Summer ESIP meeting, the cluster held a 1-day workshop on sUAS data where individual researchers and representatives from multiple commercial (Esri, DJI, SenseFly, OGC), and federal (NASA, NIST, NOAA, USGS) organizations attended and presented on their perspectives on sUAS data management approaches [40].

A key outcome of this workshop that spurred further conversations with industry, was input from sUAS users indicating that key metadata were missing from what was exposed by commercial sUAS providers. This lack limited either their ability to perform accurate analyses or to publish the data in a manner that would now be considered necessary to meet the conceptual requirements of FAIR. Some examples of these missing metadata for instance included: attitude, air speed, temperature, camera calibration date and method, autopilot firmware version, GPS instantaneous error, and many other parameters the relevance of which was use case dependant. When approached about the need for these metadata, both DJI and Sensefly responded with enthusiasm for hearing from the scientific community regarding what specific values were desired. This places the onus on the user community

to clearly articulate and clarify what metadata are desired so that commercial providers may expose such in a manner that does not divulge proprietary information.

### 2.1.2. Research Data Alliance

To engage a more global community (ESIP is a largely North American based organization), in late 2016 a sUAS Data Interest Group (IG) [41] was chartered within the Research Data Alliance (RDA). RDA is a multinational organization funded to *"...build the social and technical bridges that enable open sharing of data"* [42]. Since review and endorsement, the IG has held sessions at each of the biannual RDA plenaries; through these efforts, it has been possible to initiate working relationships with multiple other RDA groups pioneering technological, legal, political, and ethical efforts in the global push for better open data practices and tooling. Further, as an international organization RDA holds biannual meetings in 2 global hemispheres, it has been possible to engage with a geographically far larger distribution of researchers.

### 2.2. Additional Key Events and Communities

Through and beyond the RDA and ESIP, this effort has been bolstered by engagement with multiple groups specifically examining issues related to sUAS data. In many cases these groups are creating resources of value to the broader community, while others are exemplars for the sUAS community to look to for guidance and foundations. The following seeks to highlight some of these for two reasons: (1) to facilitate greater collaboration within and across domains where groups have developed a resources others might reuse and build on, and (2) to propose possible foundational building blocks from existing analogous efforts. It should be noted, however, that this list is not a complete set of all relevant parties, and is biased by (a) the practical limitations of who the volunteer-based ESIP and RDA efforts were able to reach, and (b) the reality that in many cases those doing notable work do not currently have any public facing instance of such. Regardless of these limitations, Figure 3 summarizes which organizations and community groups have been key in identifying particular challenges to sUAS data management, and the following sections briefly highlight some of the key community groups for reference and further engagement.

### 2.2.1. Oceanographic Sciences

Underwater gliders are an remarkably analogous system to sUAS, and the oceanographic research community has put significant effort into standardising their data management procedures. It may consequently serve the sUAS community well to adopt some of their tools and practices. Key members of this effort include the US Integrated Ocean Observing System (IOOS) who have a glider Data Assembly Center (DAC) [43] and have therefore defined a NetCDF standard to which glider data submitted to their data archive must adhere. The UK Oceanids command and control data system [44,45], alternatively, have a real-time web portal interface to deployed science gliders. The tool stack created to support this interface was built to enable the automation of both operations and science data analytics (including data quality control and assurance processes) and is built largely on standards by the Open Geospatial Consortium (OGC) and World Wide Web Consortium (W3C).

### 2.2.2. Atmospheric Science

In February 2017 NCAR's Earth Observing Laboratory (EOL) hosted a workshop that we participated in, with the proposed goal of: *"...to collect information about the needs of the NSF funded community in using sUAS for atmospheric research..."*. While the workshop was focused on key issues other than data management, the final report [46] emphasizes the need for formal sensor qualification research and the creation of standardized use procedures, an issue the International Society for Atmospheric Research using Remotely piloted Aircraft (ISARRA) [47] is also discussing. For instance, the impact of placement of common atmospheric sensors on multi-rotors on data quality has now been the subject of multiple studies [48–51].

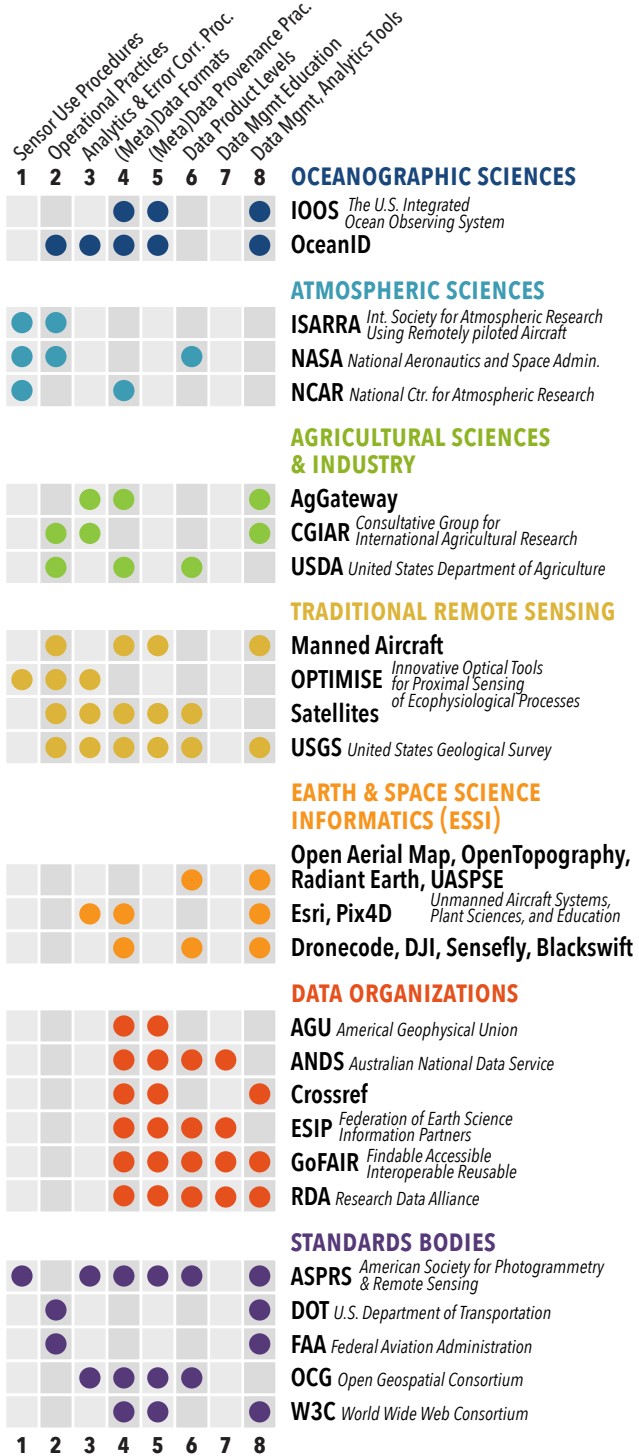

**Figure 3.** The above diagram is a summarized view of the many different communities the authors have interacted with in considering sUAS data management. Additionally, it calls out the eight key challenges to sUAS data management that this paper highlights. These eight challenges are discussed in detail in Section 3, but are listed here in order to identify primary sources for engagement. Each community, organization, and field listed here has called out these challenges through reports, papers, posters, conference sessions, community calls, and a multitude of informal conversations at various meetings, flying fields, and hallways. The grouping by discipline or role is an indication of the context within which this effort engaged with each; but in many cases these communities and organizations are contributing to multiple fields and act in multiple roles.

### 2.2.3. Agricultural Sciences

The commercial field of precision agriculture is actively investigating solutions particularly for data analytics and integration from not just sUAS but also the many diverse sensor streams now feeding commercial precision farming. An AgGateway 2017 meeting we engaged with highlighted these challenges in a panel, and commercial sUAS providers such as Sensefly are working with Esri and Pix4D analytics tool providers to develop *de facto* standards for image metadata [52]. This work includes considering mechanisms for embedding parameters regarding crop type, and status, and also factors such as current weather. Beyond these commercial efforts, the United States Department of Agriculture is also exploring standardized protocols, vocabularies, and metadata schema for sUAS data capture [53,54].

### 2.2.4. Traditional Remote Sensing

The infrastructure built to support remote sensing data management prior to the advent of sUAS (namely primarily: satellites, and manned aircraft, along with smaller blimps, kites, rockets, and balloons) are unfortunately not entirely directly portable to sUAS applications for various reasons as will be discussed in Section 3. However, while many studies continue to explore where sUAS fit within the optimal uses cases for all possible remote sensing platforms; a great deal of existing expertise, knowledge, and infrastructure can be drawn on in building new infrastructure for sUAS data. Two clear instances of this include the use of Photogrammetric techniques in stitching sUAS imagery, and the use of standardized spectral band processing algorithms and indices for sUAS data interpretation. An example of specific knowledge transfer from remote sensing to sUAS is the work by the EU based OPTIMISE [55] who have been working on standardized spectral information systems for many years, and who have most recently expanded to include practices for sUAS mounted spectral sensors. Their engagement with the spectral sensing community, including an in-depth survey of optical sUAS practices and community knowledge is ongoing, with initial survey results available in [56]. Similarly the United States Geological Survey, who have extensive experience using manned aircraft, have one of the few publicly accessible sUAS data management plans [57] based largely on their historical experience and domain knowledge.

Finally, the ESA and NASA's culture of making data appropriately open [58,59] and of using or publishing open source software [60,61] are arguably models for sUAS to follow. While manned aircraft engineering standards are not often directly applicable to sUAS, the development processes, decision metrics, and operational practices used are increasingly applicable as sUAS are integrated into controlled airspaces. Furthermore, manned aircraft data processing tool stacks are often built on widely used standards such as those from the OGC's Aviation Domain Working Group [62]; again sUAS would likely do well to follow this example.

### 2.2.5. Earth & Space Science Informatics (ESSI)

As a field of commerce and research ESSI spans effectively all of the organizations and groups identified in Figure 3. The organizations particularly named here are specifically grouped as organizations or projects who are currently working particularly on the practicalities of embedding metadata and hosting, serving, and providing cloud analytics services for sUAS data. However, these are not the only organizations working in this space as other passages in this section indicate.

Open Aerial Map [63] is a webportal for sUAS data that has arisen out of the non-governmental sector, largely driven by citizen disaster response use of sUAS. As such their focus is visible band imagery from non-scientific sUAS platforms. OpenTopography [64] provides DOIs and a searchable interface to community contributed data topography data. It is not solely focused on sUAS data is focused on high resolution data, largely but not exclusively captured using LIDAR. Data sets contributed consist of source imagery (for photogrammetry), point clouds, and raster derivatives. Radiant Earth [65] is a large scale project seeking to become a global portal and cloud analytics

provider of remote sensing data. sUAS data are not a focus but some community members have experimented with contributing visible imagery to their systems and using their tools for analytics. UASPSE (Unmanned Plant Sciences, and Education) [66] project is a National Science Foundation funded project focused on sUAS data for Agricultural research and supporting community access to such data.

All four of these portals are providing a service to a community that can be built on. The common biggest weakness across these is that while some support the submission of metadata none provide any guidance on what is required or suitable, and due to the lack of community formalization these metadata are largely not machine readable. The need for common practices regarding embedding metadata into sUAS data particularly near the source has been the subject of many conversations within this engagement effort. This has covered both the need for common parameters and schema, and available technical approaches with various commercial drone and software providers. Some of these key organizations engaged with were: the open source Linux Foundation managed Dronecode project [67], DJI, Sensefly, Black Swift Technologies [68], and as described above we have followed the work of Esri and Pix4D in developing standards for use within their tool-sets.

## 3. Results

Emerging from the above described engagement efforts have been two key results: (1) how sUAS data are unique and consequently in part require a measure of custom data management solutions, and (2) eight challenges that would need to be addressed in order for the full potential value of sUAS data to be accessed. As described, these results are not the outcomes of a formalized study or survey but are what have emerged from the focused four year effort to engage with the relevant community members (shown in Figure 2), and summarized in Figure 3.

### 3.1. sUAS Data Are Unique and in Need of Unique Management Infrastructure

**sUAS data are uniquely 5+ dimensional**

All sUAS data are associated with a location in both time and 3-dimensional space. While location- and time- stamped data are not unusual, multiple streams of simultaneously recorded values captured from a moving 3-dimensional trajectory at sporadic temporal intervals are uncommon, and this is what sUAS enable. Furthermore, to correctly interpret many sUAS data requires additional metadata, such as the time-series stream of the sUAS attitude, or an instantaneous measure of local luminosity. sUAS data is therefore unique for its mandatory 5+ dimensionality: multiple co-captured geospatially-tagged measurements of varying precision, taken within multiple discretized time periods, along a 3-dimensional trajectory.

**sUAS data provide uniquely high spatiotemporal resolutions**

sUAS are being used in the sciences largely as they are a low-cost way of quickly capturing high spatial and temporal resolution data. For instance, spatially, even low cost sUAS can achieve <5 cm/pixel horizontal ground resolution imagery, and they have the entirely unique ability to sample at similar resolutions in fully customized vertical profiles. Further, temporally, sUAS systems may be deployed both repeatedly, and dynamically in response to real-time changing circumstances, with periodicities ranging from minutes to years. This high temporal resolution is most visibly advantageous in the use of sUAS in disaster response (e.g., wildfires, flooding, or earthquakes), but it is equally useful in scientific research that can be subject to both unforeseen changes in long planned observations (e.g., unpredictable wildlife activity, or unforeseen operational restrictions) and spontaneous opportunities (e.g., an unanticipated flooding event of an area of interest). sUAS consequently are providing a uniquely high resolution low cost offering that neither manned aircraft systems or satellites—both of which require months of planning and very large budgets—nor ground based sensors or other low altitude platforms (e.g., kites, balloons) can

readily offer.

**sUAS data are classically *Big***

　　sUAS data are big in all four of the classic *Big Data* characteristic 'Vs' [69]. The *variety* in form, function and *veracity* of sUAS data is only limited by current sensor miniaturising technology and regulations, but currently commonly includes both low cost and professional grade: multi- and hyper- spectral imagery; multiple LIDAR and RADAR sensing technologies; a wide range of gas and particulate matter sensors; mechanisms for water, genomics, and other physical sample capture; and common time series parameters such as temperature, pressure, humidity, and the local characteristics of radio frequency signals. The *volume* of data that sUAS can quickly capture is nontrivial, particularly with spectral sensors, with a single flight able to return tens of gigabytes of raw data. In many cases, increasing the sampling rate at which sUAS mounted sensors can capture quality data means air speeds may also be increased allowing larger areas to be covered. It is probable that this will drive sensor engineering and consequently capture rates are likely to continue increasing going forwards as technology improves. Between increasing capture rates and growth in the use of sUAS for data capture, both instantaneous and net sUAS data *velocities* will most likely increase in the future.

**sUAS data are increasingly created by small science**

　　Large unmanned system technologies such as unmanned planes or underwater gliders have historically been accessible only to researchers working at large scale and often government based research institutions with the resources to build and maintain large scale research facilities. However, small sUAS have made it possible for small and modestly funded teams of researchers to use unmanned technologies. The adoption of sUAS technology by these smaller and more ad hoc teams has consequences for the management of these data both as it increases the quantity of data being captured by researchers overall, and because it increases the need for common practices that cross discipline boundaries. Whereas large scale research endeavors (sometimes called *big science*) often have correspondingly robust plans and infrastructures for data archiving and management, smaller scale teams (sometimes called *small science* or *little science*) have correspondingly ad hoc and idiosyncratic data management practices [11,13].

*3.2. Eight Community Distilled sUAS Data Management Challenges to Be Addressed*

1. **Sensor use procedures:** Sensor specific, tested and qualified use procedural best practices and standards are urgently needed in common human and machine readable languages. These best practice methodology and procedural guidelines should be developed and provided either by the manufacturer or the research community and include: mounting requirements on various platforms, calibration, ground truthing, and maintenance procedures, sample rates, flight patterns, and required metadata for data use and publication. The need for these and the aforementioned emphasis on machine and human readability is both for user ease and so as to enable greater automation in the capture of data provenance. As mentioned existing initial work on this issue has already appeared within the atmospheric community [48,49] and the Agricultural Sciences [52]. While these procedures are largely currently not instantiated in open machine readable forms, they represent a direction for others to follow and contribute further to.

2. **Operational practices:** Having best practices regarding operational protocols for scientific research will lower the barrier to entry for new users, allow training materials to be standardized for the many new training courses being created, and reduce the burden on operators which can only lead to safer operations. Further, while many countries have now begun to settle on regulations, many research organizations are still grappling with their own internal policies and protocols. Researcher operational best practices, created based on the experience of those who have been operating for longer, could serve to accelerate organizational protocol deployment in a country agnostic manner. One examples of such that is readily accessible comes from University

　　　of Exeter's Remote Sensing Laboratory [70], and another is the University of California's risk
　　　assessment and operating policy [71].

3. **Analytics and Error correction procedures:** Best practices and acceptable error tolerances for primary sensor taxonomy branches and the associated processes need to be defined so as to avoid unintentional—but easy to introduce—errors [72]. These are needed equally by tool providers (commercial and open source) so as to allow them to build to a standard, and by user community so as ensure correct data interpretation. Defining such will additionally contribute to efforts to define sensor use best practices and metadata creation, capture, and archive tooling.

4. **Data and metadata data formats:** Guidelines regarding best practice metadata and data formats would serve the community, not as any form of restriction, but rather as a simple means of reducing workloads for both research sUAS operators and technical developers of: sensors, sUAS platforms, and the many components necessary in a data management tool stack. Having published recommended open formats based on community experience would similarly lower the barrier to novel experiments and enable both open source and commercial developers to create reusable tools.

5. **Data and metadata provenance practices:** Given that a typical sUAS data capture project involves multiple: sensors, mechanical and electrical platforms, complex data transformations, and stakeholders, and that information regarding each of these commonly has a bearing on how a dataset should be interpreted. The provenance and workflow metadata—the record of the processes that created the data—are particularly important. Definitions of what parameters are required to make a data value, set, or product reusable—in potentially other scenarios than that for which it was originally captured or created—is necessary as both a practical guideline for operations and to facilitate the creation of tools to support the automated capture of this provenance.

6. **Data product levels:** Defining suggested data product levels for various data types would facilitate both data archives and single researchers in determining what data should be archived, at what quality levels, at what resolutions, and with what associated metadata as required for likely reuse. This could be done for various primary parameter taxonomy branches, such for spectral data captured for Agricultural Sciences, and for atmospheric time series for Atmospheric Sciences.

　　　A crucial and complex sub-component to data product level definitions is the potential ethics driven policies that will govern sharing sUAS data. FAIR does not require open access, and others are exploring the ethical implications of both FAIR and open data in general [73,74]. Not least because of their historical military associations of sUAS but also due to the potential to easily violate important privacy restrictions with sUAS mounted sensors, the community needs to discuss both locally and internationally, what best practices might be for governing sUAS data's desirable degree and form of openness.

7. **Data management and analytics tools:** As shown in Figure 3, many of the relevant organizations already have some portion of a sUAS data analytics and management tool stack. However, the tools these bodies offer are only sUAS specific in a minority of cases. Rather, the majority were developed for other data types and are now being adapted for sUAS. More resources and effort are therefore necessary to accelerate these adaptations; and it is noteworthy that by addressing the above challenges, it would becomes significantly easier for resource pooling across development efforts.

8. **Data management education:** As the domain grows there is an increasing demand for introductory information that properly addresses the multitude of new expertise needed to effectively use sUAS. In response many universities and other institutions are beginning to formally train research sUAS operators. An acknowledged but core missing component of these training curricula is any information on comprehensive consideration for science data good

practices. Bringing together data management training and sUAS training offers a convenient opportunity, but one that depends heavily on investment being made first in the above challenges.

## 4. Discussion

As detailed in Section 3, the primary outcomes of engaging with the nascent sUAS community are: (a) the identification of how sUAS data are unique and where there are shared characteristics with more traditional data capture platforms, and (b) that as a result there are eight community identified challenges to improving sUAS data management.

Regarding how sUAS data are unique, the following should be noted. (1) Though there are many geospatial data formats that capture vector and raster data, stationary time series data, and high dimensionality data, and while tool stacks exist for processing and managing these data, these tools do not currently readily support the particular combination of metadata streams and multiple parameter capture sUAS data often consist of or require for correct interpretation. Similarly (2), the high spatial and temporal resolutions sUAS data are capable of capturing presents a new complicating factor for data management infrastructure. These resolutions require both potentially new multidimensional formats, schemas, and ontologies (or at least new workflow tools for handling the novel combination of such sUAS data involved), and also demand high processing times, more automated quality control, and new data product distribution tools. Considering (3) that the majority of tool stacks build for *BigData* assume operation on cloud or at least mains powered computing resources, in many cases there is a need for real-time sUAS data processing on low power or low bandwidth edge compute devices. Finally (4), research has shown [14,75,76], that the range of data practices utilized by smaller teams should be considered a feature rather than a *bug*; this is because the data workflows and practices must be customized to the unique contexts and goals of a given group, project, and organizational structure. Standardized workflows across all smaller research teams are neither achievable nor desirable. Consequently, sUAS data management solutions need to be created with the necessarily diverse data practices of a small lab researcher specifically in mind, and this is all the more so true given the wide spectrum of disciplines sUAS users include.

Regarding the challenges outlined. As new sensors, sUAS platforms, and analytics techniques develop, it is clear that addressing solutions to these challenges will require updates and extensions. However, initial efforts on each are the only way to ensure such periodic updates, extensions, and community-driven maintenance will be plausibly practical, sustainable, and backwards compatible to any degree in the long term. Further, by initiating the development of solutions to any of the following in a collaborative manner with a view to long term sustainability, partial solutions will be both immediately accessible for use by others and accessible for extension, iteration, and improvement such that gradually more complete solutions naturally arise. That is, provided long term maintainability and extensibility are considered in initial work.

## 5. Conclusions

The use of sUAS for data capture is increasing rapidly, both for commercial applications and as a new platform for data capture for a wide and diverse spectrum of research fields. As a nascent field with many avenues of development underway to increase both operational and scientific platform maturity, the issue of managing and optimising the data flow from sample to knowledge product has not been extensively explored. This paper describes an effort to explore what resources are currently available for handling sUAS data, what approaches are currently being used, and where there are challenges to fully realising sUAS data's value. As a largely unfunded effort subject largely to the authors abilities to take advantage of opportunities that either arose organically or were commensal, this exploration was not comprehensive. It has, however, engaged with a significant breadth of domain users, developers, commercial participants, and analogous mature fields from which sUAS might learn. In addition to finite scope, a key limitation in this engagement is that the majority of work was done in North America, however, this was not exclusive, with 6 out of the 28 formal engagements

listed occurring elsewhere in the world. To the best of our knowledge this is the only effort to achieve the above at any international scale.

There are two significant novel outcomes of this work. (1) The identification of the combination of characteristics that sUAS data commonly exhibit, shows that while it shares many traits with more traditional methods of data capture, the combined differences mean existing infrastructure as currently developed and deployed is not capable of enabling users to fully realize the potential value of sUAS data. These primary characteristics were: (i) sUAS data are uniquely 5+ dimensional, (ii) sUAS data provide uniquely high spatiotemporal resolutions, (iii) sUAS data are classically *Big*, and (iv) sUAS data are increasingly created by small science. (2) The detailing of eight specific challenges that must be addressed in order for sUAS to become a trusted, reliable, and optimally useful data capture platform: (i) Sensor use procedures, (ii) Operational practices, (iii) Analytics and Error correction procedures, (iv) Data and metadata data formats, (v) Data and metadata provenance practices, (vi) Data product levels, (vii) Data management and analytics tools, (viii) Data management education.

Based on these, we conclude that a conscious and determined effort by a global selection of researchers, to openly draft community driven data management best practices for the capture and management of sUAS data, would likely realize many gains and be an important step towards supporting the reproducibility and reliability of drone data research, as well as increasing the reuse of sUAS data. In the immediate future, it would cost time and effort, but in the very near future it would; (i) significantly reduce the total quantity of poorly curated sUAS data likely to otherwise be lost in the near future; (ii) minimize the length of what will otherwise be an extended period of partial and inadequate data management tooling for sUAS users making operations inefficient over a longer period of time than necessary; (iii) allow the community to circumvent the familiar larger and more expensive challenges of legacy data rescue and community wide retooling, and retraining; (iv) lower the barrier to entry for researchers entering the field and seeking to produce robust and reusable data; (v) enable collaborative rather than disparate and ad hoc building of common sUAS data infrastructure; and finally (vi) increase the transparency of sUAS data processing workflows.

The window of opportunity within which to craft such is finite and closing, given the immediate need for data tooling and practices and already growing set of sUAS data. Two possible relatively simple future tasks that may serve the community well as initial steps towards addressing these challenges include: a comprehensive and detailed review and comparison of what metadata are exposed on different common sUAS platforms, and a formal survey of existing sUAS data management approaches and different data analytics algorithms used.

**Author Contributions:** J.W. and L.B. have led the effort behind the research and writing required to produce this article. A.T. has been intrumental in contributing to the framing of these ideas within the broader context of information science. J.A., D.S., C.C., C.P., J.K., S.R.S., and T.B. were all leading collaborators in gathering the information and concepts presented here both in the article writing process and over the course of the work investigative.

**Funding:** This research was partially funded by Earth Science Information Partners and The Research Data Alliance US.

**Conflicts of Interest:** The authors declare no conflicts of interest.

## Abbreviations

The following abbreviations are used in this manuscript:

| | |
|---|---|
| sUAS | Small Unmanned Aircraft Systems |
| IG | Interest Group |
| NASA | The National Aeronautics and Space Administration |
| USGS | United States Geological Survey |
| USDA | United States Department of Agriculture |

# Appendix A

**Table A1.** References for Figure 2.

| Notes | Speaker | Link |
|:---:|---|---|
| 1 | L Barbieri, J Wyngaard | http://commons.esipfed.org/2015SummerMeeting |
| 2 | J Wyngaard, L Barbieri | http://commons.esipfed.org/node/8799 |
| 3 | M. Messinger, R. Hunt, L. Barbieri | http://commons.esipfed.org/node/8798 |
| 4 | T Corbett, L Barbieri, S Barberie | https://2016esipsummermeeting.sched.com/event/6uH7/drones-explore-the-landscape-technical-physical |
| 5 | K Bhakta, W Teng | https://osf.io/ub84e/ |
| 6 | W Teng | https://2017esipwintermeeting.sched.com/event/9BKT/uassdrones-in-agriculture |
| 7 | K Edmonds, W Teng | https://www.lucidchart.com/documents/view/c99183b1-658a-4b36-8848-490b89344800/0 |
| 8 | Multiple speakers | https://osf.io/view/esipsummermeeting2017/ |
| 9 | A Thomer, L Barbieri, J Wyngaard | https://www.esipfed.org/meetings/upcoming-meetings/esip-winter-meeting-2018 |
| 10 | J Wyngaard, A Thomer, L Barbieri | https://digitalag.org/unmanning_data.php |
| 11 | A Thomer, J Wyngaard | https://www.esipfed.org/meetings/upcoming-meetings/esip-winter-meeting-2018 |
| 12 | L Barbieri | http://www.aggateway.org/eConnectivityActivities/Councils/PrecisionAg.aspx |
| 13 | J Wyngaard | http://optimise.dcs.aber.ac.uk/workshops-meetings/2017-3/annual-workshop-and-mc-meeting-february-2016/ |
| 14 | L Barbieri | https://ams.confex.com/ams/97Annual/webprogram/Paper317061.html |
| 15 | L Barbieri, J Wyngaard | https://agu.confex.com/agu/fm15/meetingapp.cgi/Paper/80038,https://agu.confex.com/agu/fm15/meetingapp.cgi/Paper/84585,https://agu.confex.com/agu/fm15/meetingapp.cgi/Paper/69625 |
| 16 | L Barbieri, J Wyngaard | https://agu.confex.com/agu/fm16/meetingapp.cgi/Paper/175599 |
| 17 | L Barbieri, J Wyngaard | https://agu.confex.com/agu/fm17/meetingapp.cgi/Session/30530 |
| 18 | L Barbieri | https://agu.confex.com/agu/fm18/meetingapp.cgi/Paper/349789 |
| 19 | J Wyngaard, L Barbieri | http://www.codata.org/events/conferences/international-data-week-2016 |
| 20 | L Barbieri, J Wyngaard | https://www.rd-alliance.org/groups/small-unmanned-aircraft-systems%E2%80%99-data-ig,https://www.rd-alliance.org/blogs/drones-emerging-scientific-tools-trade.html |
| 21 | L Barbieri | https://www.rd-alliance.org/ig-small-unmanned-aircraft-systems-data-rda-10th-plenary-meeting |
| 22 | J Wyngaard, K Anderson | https://www.rd-alliance.org/plenaries/rda-eleventh-plenary-meeting-berlin-germany/rda-11th-plenary-programme |
| 23 | J Wyngaard, T Motshegwa | https://www.eventbrite.ca/e/drones4good-international-data-week-tickets-51720620769 |
| 24 | J Wyngaard, L Barbieri | https://rd-alliance.org/ig-small-unmanned-aircraft-systems%E2%80%99-data-rda-12th-plenary-meeting |

**Table A1.** *Cont.*

| Notes | Speaker | Link |
|---|---|---|
| 25 | J Wyngaard, L Barbieri, J Klump, T Bell, T Desell | https://drive.google.com/drive/folders/1EU1cyfLAMPXw PhpdPL7Jn-Exn7Jyus1W |
| 26 | L Barbierii | https://www.eol.ucar.edu/system/files/UAS.Workshop.20 180206.pdf |
| 27 | C Vardeman, A Thomer, J Adams, L Barbieri, J Wyngaard | https://github.com/Vocamp/dronedata |
| 28 | J Wyngaard | http://www.c3dis.com/ |
| 29 | L Barbieri | http://www.isarra.org/?page_id=377 |

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
