# Peer review of "Emergent Challenges for Science sUAS Data Management: Fairness through Community Engagement and Best Practices Development"

_remotesensing, doi:10.3390/rs11151797_

Round 1

Reviewer 1 Report

The authors have summarized the current data management problems on sUAS data collection. This paper is a good summary document to raise up those challenges. In general, the paper is well written and suitable for publication. However, as commented in the attached PDF file, minor editing work is required for clarification.

Author Response

No corrections suggested in commentary review.

Pdf comments corrected:

We merged Figure 1&2 so as to easily present them on 1 page as advised.  Additionally the error in footnote generation has been corrected

Mis-capitalisation of “ESRI” has been corrected thrice

It is suggested that “To be addressed” in line 290 be replaced with “described”.  But this would mean the root “describe” is used in 2 consecutive sentences.  To avoid this the original wording is retained.

Line 290 “describe” is corrected to “described”

Line 294 “4 year” is corrected to four-year

Line 296 ‘is’ has been corrected to ‘are’

Line 326 GB corrected to “gigabytes”

Line 401 “Figure” is added

Line 404, space removed

Line 405 “noteworthy” corrected

Line 410 space removed

The following texts were highlighted by the reviewer but no comment was left and the intended critique or desired correction is unclear.

Line 90 “date” 

Line 227  “studies[49–52]).”

Reviewer 2 Report

See comments in attached.  Comments and grammar elements are contained therein.

Author Response

Does the introduction provide sufficient background and include all relevant references?: 

    Reviewer: “Can be improved”

    Response: We have reviewed and revised the section with the goal of clarifying the manuscript and including sufficient references.

Are the results clearly presented?

    Reviewer: ”Can be improved”

    Response: We have reviewed and revised the section with the goal of clarifying the manuscript.

Pdf comments corrected:

Line 54 “mean” corrected to “means”

Line 63 “Parameter-rich” concatenated

Line 64 “high-resolution” concatenated and coma removed

Line 68:  Reviewer advised re-wording the following considering FAA Part107 attrition rates data: “And this growth is more than matched by the commercial sUAS sector”

Response:  While we do not doubt the reviewer's comment we were unable to find this or similar data.  The citations given in the paper as motivation for this sentence are 2018 and 2019 business forecasts that all indicate continued rapid growth of the sUAS market.  Such market forecasts are by definition approximations and predictions that have inherent inaccuracies.  However, the authors were unable to find equally notable data or forecasts indicating a significant slow down or shrinkage in the use of drones.  To be more clear about this uncertainty the sentenced was changed to the following:

“This growth is paralleled by significant growth of the commercial \uas sector, with some forecasts estimating a market value of USD 100 billion in the next five to ten years [5–8].”

Line 77: “Purpose-built” concatenated

 Line79: “post-processing” concatenated

Line 113: Reviewer queried the sentence : “Without shared data practices and methods….”  stating “what about the practices and standards developed by ASTM, AUVSI TOP, ANSI etc?”

As is detailed later in the paper, we do not intend to claim that there are no relevant standards specific to drones.  But rather that those available (such as ASTM, AUVSI TOP, and ANSI, or for instance the netCDF, ASPRS, OGC, Esri and other standards named in the paper) are either or both: useful but insufficient, or not applicable to the particular use case this paper is describing, namely specifically the need to capture the metadata required to make drone captured data broadly FAIR.  We think the full sentence, within the context of the complete paper, is sufficiently justified and correct as it is.

Line 114: “data-based” concatenation

Line 132: “ cross-disciplinary” concatenation

Line 157: “Volunteer-based” concatenation

Line 165:  Reviewer queries if the fact that the process was driven by community input was a limiting factor given that the communities listed in Figure 2.1  “seems AG heavy”.

The authors wish to respond that the intention of this sentence was indeed to highlight that given the breadth of relevant fields, indeed the engaged community is in effect a limiting factor.  However, the communities engaged with as depicted by Figure 2.1 is very much not “Agriculture heavy”.  Rather the ESIP and RDA meetings that dominate the list are respectively Earth Science and Academic research focused foremostly, but both intentionally cover a much wider spectrum of fields. Further, other organisations and communities that specialise in other fields were consulted multiple times.  To clarify this point the highlighted sentence has been elaborated on as follows: 

“This process was largely driven by academic researcher needs, the perceived value opportunity, and community request and interest.  As a result the focus has been on academic data more than commercial, however, both were consulted and across both the spectrum of fields of expertise engaged spans Engineering disciplines, Earth and Environmental Sciences,  Agricultural Sciences, and the Humanities”

Line 211: adapt changed to adopt

Line 215: “real-time” concatenation

Section 2.2.3 has the note: “Does there need to be an explanation of the relationship of Exif format standards.”  This is presumably in reference to the citation of Pix4D’s metadata standards in this section.  Specifically the sentence: :....with Esri and Pix4D analytics tool providers to develop de facto standards for image metadata [53]”.  The authors are inclined to answer no as they have not elaborated on what a csv or NetCDF or other format is elsewhere in the paper and as the sentence assumed to be applicable doesn’t name “exif” but uses a citation that does when referring to image metadata.  It should be easily inferable therefrom that the citation’s reference to Exif is “image metadata”

Page 10 has a note question the formatting and suspecting it to be an artifact of the pdf generation.  This is indeed the case and is fixed.

Line 286: “tool-sets” concatenated

Line 290: Fixed “described”

Line 290: Reviewer indicates a perceived discrepancy between the abstract and this sentence.   It’s not clear exactly what the discrepancy (beyond type setting and grammar) is.  

Line 405: Noteworthy

Line 476: “Community-driven” concatenation

Line 484: “Community-wide” concatenation 

The following texts were highlighted by the reviewer but no comment was left and the intended critique or desired correction is unclear.

Line 33  “:“

Line 59 “lowered risks to”  //Comment given appears to be empty…?

Line 319: ‘V’s’

Line 334: “historically been accessible only to researchers working at large scale and often government based research institutions with the resources to build and maintain large scale research facilities.”

Reviewer 3 Report

This manuscript presents the characteristics of UAV data and practices of surmounting the challenges of UAS data management. By investigating a long term UAS data acquisition and management plans, authors elaborate the available resources for dealing with UAS data, comparison of traditional approaches, and challenges to realize the full value of UAS data. The manuscript is in general well-written and easy to follow. However, I do have some significant concerns before it accepted.

Major concern:

The review of current UAV data management methods is superficial, more detailed examples on data management approaches are needed. It would be more important to address which raster data management functions in some wide used platforms are in need to better manage the UAS data.

Specific comments:

(1)   Section 2.2, it is not necessary to spend a huge amount of text in introducing the data and meta data. Authors need to discuss how the specific data properties afford the additional challenge on the data management.

(2)   Authors mentioned both Esri and Pix4D were developing standards for UAS data management within their software tools. More analysis and comparison are needed for their data management platform, e.g. Drone2Map with ArcGIS online and Pix4D cloud. Both of them integrated the cloud and local data management functions in processing.

(3)   An extended review on UAS data filtering, assimilation, and blending are also worth to discuss, which is also a part of data management approach.

(4)   Section 2.2.4, the section title is too general in my opinion.

Author Response

Are the methods adequately described? 

Reviewer: Can be improved

Response: We have reviewed and revised the section with the goal of clarifying the manuscript.

Are the results clearly presented? 

Reviewer: Can be improved

Response: We have reviewed and revised the section with the goal of clarifying the manuscript.

Are the conclusions supported by the results? 

Reviewer: Can be improved

ResponseWe have reviewed and revised the section  with the goal of clarifying the manuscript.

Reviewer 3 additionally raised the following critiques:

1) “The review of current UAV data management methods is superficial, more detailed examples on data management approaches are needed. It would be more important to address which raster data management functions in some wide used platforms are in need to better manage the UAS data.”

The authors agree that an in-depth review and comparison of key commercial sUAS platforms and service providers regarding: which metadata are exposed to users, which underlying algorithms are used in any data processing, and what services features are provided by each,   would be of notable value to the user community.  For instance, the authors are aware that Esri has several possible solutions for producing authoritative mapping products from drone imagery, each targeted for different users:  (1) Drone2Map for ArcGIS, (2) the ortho mapping capability of ArcGIS Pro Advanced, and (3) the Ortho Maker app included with ArcGIS Enterprise. Each of these solutions serves a different role in a user’s data management toolchain.   

However, this paper was not intended as a review or critique of specific products.   Rather it was through our engagement with many sUAS users that we determined there were key metadata missing from what was exposed by commercial providers and that this lack limited either their ability to perform accurate analyses or to publish the data in a manner that would see them meet the conceptual requirements of FAIR.  Some examples of these missing metadata for instance included: attitude, air speed, temperature, camera calibration date and method, autopilot firmware version, GPS instantaneous error, and many other parameters the relevance of which was use case dependant.  When approached about the need for these metadata, overwhelmingly all commercial drone platform and service providers responded with enthusiasm for hearing from the scientific community regarding what specific values were desired.   This places the onus on the user community to clearly articulate and clarify what metadata are desired so that commercial providers can expose such in a manner that does not divulge anything proprietary. 

While phrased differently this need is articulated in section 3.2 Challenge 5: “Data  and  metadata  provenance  practices” and the following paragraph was added to section 2.1.1

“A key outcome of this workshop that spurred further conversations with industry, was input from sUAS users indicating that key metadata were missing from what was exposed by commercial \uas providers.  This lack limited either their ability to perform accurate analyses or to publish the data in a manner that would now be considered necessary to meet the conceptual requirements of FAIR.  Some examples of these missing metadata for instance included: attitude, air speed, temperature, camera calibration date and method, autopilot firmware version, GPS instantaneous error, and many other parameters the relevance of which was use case dependant.  When approached about the need for these metadata, both DJI and Sensefly responded with enthusiasm for hearing from the scientific community regarding what specific values were desired.  This places the onus on the user community to clearly articulate and clarify what metadata are desired so that commercial providers may expose such in a manner that does not divulge proprietary information. “

2)   “[in] Section 2.2, it is not necessary to spend a huge amount of text in introducing the data and meta data [sic]. Authors need to discuss how the specific data properties afford the additional challenge on the data management.”

Section 2.2 “Additional key events and communities:” is within “Material and Methods” where for this paper the content is a summary of how the authors engaged the broader sUAS community.  Section 2.2 specifically is a summary of the responses we received and observed from core communities that share a common focus independent of sUAS.  It is unclear what text regarding introducing data and metadata the reviewer specifically refers to.   

However, within the broader context of reviewer 3’s comments, perhaps the request is for more specific examples of how sUAS data “afford the additional challenge on the data management”[sic].  In which case, the authors wish to point to section 3.1 where specifics are given as to how sUAS data exhibits unique properties and section 4 for how this therefore leads to the challenges outlined.

3) “Authors mentioned both Esri and Pix4D were developing standards for UAS data management within their software tools. More analysis and comparison are needed for their data management platform, e.g. Drone2Map with ArcGIS online and Pix4D cloud. Both of them integrated the cloud and local data management functions in processing.”

The authors again agree an analysis and comparison of different sUAS commercial data management platforms would be a valuable contribution to the community but disagree that such should be in this paper specifically.  Rather we consider this to be future work for the following reasons: (1) as a scholarly work we do not believe we should necessarily be critiquing specific commercial products, (2) the question of what metrics would be used for such a comparison is highly dependant on the use case or underlying goal, and (3) any such a specific comparison such as might focus on for instance sUAS captured image data, was not the goal of this work.  Rather, the authors sought to review, from both an application and data type agnostic view point, the state of sUAS data management and to then present what they perceived as the most urgent challenges.

 An associated and valuable potential future comparison for instance may be to compare data management tools according to how they enable or fail to enable sUAS data being published as FAIR.  The subject of FAIR metrics is currently a very active topic, and it may therefore be worth waiting for the community to come to a stronger consensus regarding these metrics perhaps before carrying out such a comparison.  

4) “An extended review on UAS data filtering, assimilation, and blending are also worth to discuss, which is also a part of data management approach.”

Again, the authors do agree such a review would indeed be of value, but disagree that this should be in this particular paper again given the goal of this paper was to provide a higher level review of sUAS data management as a whole.

5) “Section 2.2.4, the section title is too general in my opinion.”

Section 2.2.4, is within section 2.2 which attempts to summarise the general responses we received and observed from core communities that share a common focus independent of sUAS.  “Traditional Remote Sensing” was used as a broad and general term to identify remote sensing techniques that predate sUAS, namely those identified as “...satellites and manned aircraft…”.

To improve and clarify this intended meaning, the first 2 sections of section 2.2.4 have been altered to now read: “The infrastructure built to support remote sensing data management prior to the advent of sUAS (namely primarily: satellites, and manned aircraft, along with smaller blimps, kites, rockets, and balloons) are unfortunately not entirely directly portable to sUAS applications for various reasons as will be discussed in Section 3.  However, while many studies continue to explore where \uas fit within the optimal uses cases for all possible remote sensing platforms; a great deal of existing expertise, knowledge, and infrastructure can be drawn on in building new infrastructure for \uas data.”

Final collective response to points 1, 3 and 4

In addition to the above, given the common thread regarding important additional work the reviewer has highlighted, we added the following sentence to our conclusions highlighting these important themes.

“Two possible relatively simple future tasks that may serve the community well as initial steps towards addressing these challenges include: a comprehensive and detailed review and comparison of what metadata are exposed on different common sUAS platforms, and a formal survey of existing sUAS data management approaches and different data analytics algorithms used.”

Round 2

Reviewer 3 Report

Thanks for addressing my comments. Now the manuscript reads well and fit the topic of the journal.